# Summer decapod crustacean larval communities along the eastern Spanish Mediterranean coast

**Marta Carreton**[1]\*, **Guiomar Rotllant**[1], **Diego Castejón**[1,2], **Nixon Bahamón**[1], **Joan B. Company**[1]\*

1 Institut de Ciències del Mar (ICM-CSIC), Barcelona, Spain, 2 Centro Maricultura da Calheta, Direção Regional do Mar, Calheta, Madeira, Portugal

\* mcarreton@icm.csic.es, martacarreton@gmail.com (MC); batista@icm.csic.es (JBC)

**Data Availability Statement:** The data will be available at SEANOE public repository. https://doi.org/10.17882/81689.

## Abstract

Decapod crustaceans are a diverse group englobing several species of commercial and ecological interest. In the Mediterranean Sea, decapod crustacean fisheries are among the most profitable, although in many cases their early life stages are poorly known. In this study, we tackle the composition and diversity patterns of the decapod larval communities along the eastern Spanish Mediterranean coast. Zooplankton sampling was carried out in surface waters at 101 stations from July 20th to August 31st 2016, over bottom depths between 90 and 1840 m. All shrimp larvae were identified to the lowest possible taxonomical level, and larvae from Anomura, Achelata and Brachyura were left at infraorder level. No larvae of Astacidea or Polychelida were found. The total zooplankton volume was estimated. A total of 20,022 decapod crustacean larvae were identified, focusing on shrimp taxa (suborder Dendrobranchiata and infraorder Caridea). Both zooplankton volume and decapod larval density values were higher in the northern part of the studied area, cut by deep submarine canyons. After assessing the diversity parameters of the decapod larval community, we present the summer mesoscale larval distribution of several species of commercial interest such as the caramote prawn (*Penaeus kerathurus*) or the deep-water rose shrimp (*Parapenaeus longirostris*). The northern submarine canyons are dominated by the presence of Penaeoidea, being the deep-sea shrimp *Aristeus antennatus* the dominant species in the community in this area, while the Sergestoidea are more abundant in the southern zone. This is the largest-scale study on decapod larvae mesoscale distribution in the Mediterranean Sea.

## Introduction

Decapod crustaceans are a diverse group englobing several species of commercial and ecological interest [1]. In the western Mediterranean Sea, decapod crustacean fisheries are among the most profitable, accounting for over 250 million euros in annual revenues while representing less than 1% of the catches in weight [2]. The diverse geomorphological structures along the eastern Spanish Mediterranean coast allow for a wide range of decapod crustacean target species for fisheries in different habitats [3], many of which reproduce during the summer months

**Funding:** This research was carried out under project CONECTA (CTM2014-54648-C2) funded by the Spanish Miniterio de Economía y Competitividad. MC benefited from a FPU2015 grant from the Spanish Ministerio de Educación. The funders had no role in study design, data collection and analysis, decision to publish, or preparation of the manuscript.".

**Competing interests:** The authors have declared that no competing interests exist.

[4]. In the deep submarine canyons that cut the continental shelf in the northernmost area, near the French border, the trawling fleet focuses on deep-sea species that dwell relatively close to the coast, such as the deep-sea red shrimp *Aristeus antennatus* [e.g. 3, 5, 6]. The wide continental shelf off the coast of Tarragona and the Ebro Delta allow for fisheries of shallow and estuarine waters, such as *Parapenaeus longirostris* or *Penaeus kerathurus* [7]. In the southernmost area, starting at hydrodynamic barrier of the Ibiza Channel, the fleet also targets the pandalid shrimps of the genus *Plesionika* [8, 9].

The order Decapoda is divided into two suborders: the Dendrobranchiata, the most primitive group of shrimps whose females release their eggs directly into the water column, and the Pleocyemata, which includes shrimps, crabs and lobsters, and whose females bear their eggs in their abdomen. The larval stages of these groups are planktonic and many carry out diel and ontogenetic migrations of which very little is known [10–13]. Dendrobranchiata eggs hatch as a free-living, non-feeding phase called nauplius, which metamorphoses into the subsequent feeding phases called protozoea, mysis and decapodid [14]. In the case of the Pleocyemata, the eggs can hatch directly into the first zoeal stage, which undergoes several metamorphoses until it reaches the final decapodid stage that preeceds the juvenile life phase [14]. In some cases, larvae undergo abbreviated or direct development and omit some of these phases [14].

The challenging and time-consuming task of identifying decapod larvae to species level has limited the study of these communities, so much so that we can only count a few of these studies in the Western Mediterranean Sea [12, 15–20]. Currently, the descriptions of many decapod crustacean larvae are outdated or do not follow modern standards, and some larval stages of decapod taxa have yet to be found in the plankton or confirmed by molecular methods [21]. The existing studies in Spanish waters mainly focus on a limited spatial area while covering the whole decapod community [Cf. 22–24], whereas other studies choose to focus on one taxonomic group, such as brachyuran crabs [25, 26]. Molecular techniques such as DNA barcoding are growingly used to aid in the identification process of decapod crustacean larvae [27–29]. The interest of covering a wide geographical area such as the eastern Spanish Mediterranean coast lies in observing how diversity patterns change through different geomorphological structures and environmental conditions.

The water circulation in this area is dominated by the southwestward Northern Current, modified by different geomorphological and hydrodynamic structures. Previous studies have shown submarine canyons and seasonal gyres as particle retention sites [30, 31], which can affect the distribution of eggs and larvae of fish and crustaceans [32–34]. Moreover, a hydrodynamic model run in this area, designed as the fisheries management unit Geographical Subarea 6 (or GSA 6) by the General Fisheries Commission for the Mediterranean (GFCM), showed diverse circulation patterns the dispersal patterns of particles, and particularly larvae of *A. antennatus* (Fig 1; [35, 36]). The area is mainly influenced by the Northern Current [37], which has been found to transport larvae southwestward [38], whereas that its interaction with the Balearic current may generate seasonal eddies restricting the dispersal of larvae or transporting them offshore [38, 39].

The objective of this study was to describe the summer community of decapod crustacean larvae along the Eastern Spanish Mediterranean coast, and to give information on decapod species of ecological or commercial interest in the area.

## Method

### Zooplankton sampling and larvae identification

Sampling was carried out at 101 stations of GSA-6 from July 20$^{th}$ to August 31$^{st}$ 2016 on board the Spanish research vessel *García del Cid*, in the frame of coordinated project CONECTA

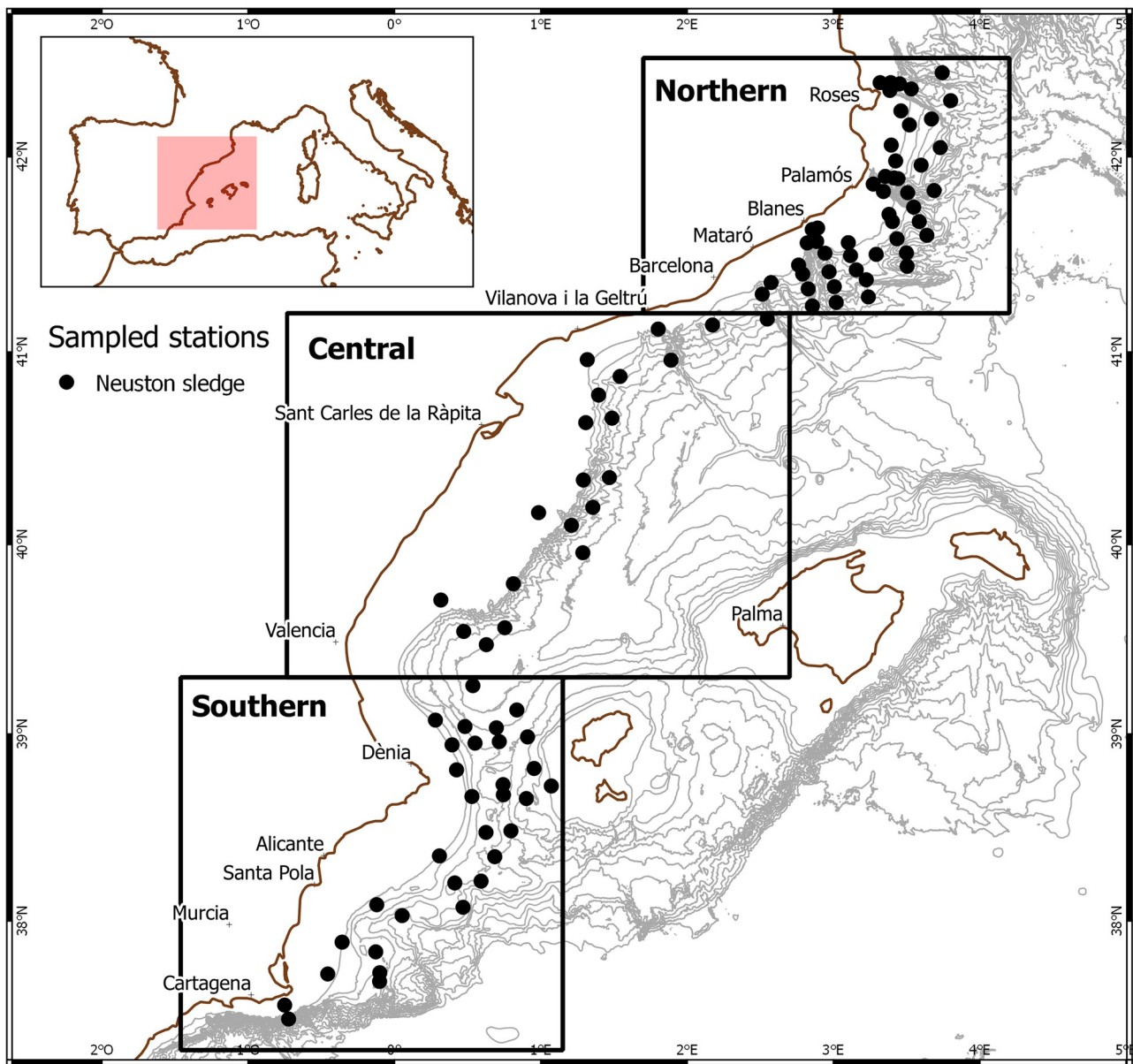

**Fig 1. Division of GSA 6 in three zones (Northern, Central and Southern) according to references in the literature [35, 36].** Black dots represent stations sampled with neuston sledge.

(CTM2014-54648-C2) (Fig 1; further information on sampling station locations can be found in S1 Table). Sampling direction was from South to North. The sampling was carried out using a 1-m wide neustonic sledge with a 300-μm mesh net, towed horizontally for 15 minutes to obtain plankton samples of the upper end of the water column (0.5–1 m). Sampling was carried out avoiding the dusk and dawn periods, with 46 stations sampled during the day and 55 at night. For the purposes of this study, day and night samples were treated together. The volume of filtered water was monitored with a flowmeter. The samples were rinsed with distilled water and stored in 96% ethanol on board.

In the laboratory, an estimation of zooplankton volume per sample was made pouring the contents of each rinsed sample into a 500 mL measuring cylinder. After a settling time of 5 minutes, the zooplankton volume was noted. All decapod crustacean larvae were sorted from the samples and identified using a Leica M205C stereomicroscope and the available literature [18, 40–43]. We aimed for the lowest taxonomic level in the case of shrimp taxa, leaving crab and lobster taxa at infraorder level (i.e., Anomura, Achelata, Brachyura). In some cases, the state of preservation of the specimens did not allow for the identification to species level, especially in the case of taxa with particularly fragile structures, such as the spines of the sergestid shrimps. Three Dendrobranchiata larvae were sent to AllGenetics facilities (A Coruña, Spain) for molecular confirmation of the identification using Cytochrome Oxidase I (COI) gene, following the method detailed by Carreton et al. [28]. The GenBank accession numbers to the resulting sequences are OP278973 to OP278975.

### Environmental data collection

Oceanographic data (water temperature, salinity and fluorescence) were collected at each station with a SBE25 and a SBE911 CTD (SeaBird Electronics). A post-cruise inter-calibration among CTDs was carried out to make the data comparable. Due to the long time span of the sampling cruise, averages from daily means from July 20th to August 31st were calculated for each variable.

### Data analysis

The larval abundance data were standardised to individuals per 1000 $m^3$ and logtransformed ($log_{10}$ + 1) to adjust a linear model and assess correlation of total decapod larval density with zooplankton estimated volume. Individual species abundance maps were built with QGIS [44] using inverse distance weighting interpolation. Species richness, Shannon diversity index and Pielou evenness index were calculated for each station, as well as the average abundance and frequency of occurrence (in %) for each species and taxonomical group. For the spatial analysis, the studied area (GSA6) was divided into three zones according to environmental characteristics such as temperature, salinity, bathymetry and surface circulation, influencing the larval dispersal, as mentioned in previous studies in the area [35, 45]. The Northern zone englobes the deep submarine canyons from the Gulf of Lions to the coast of Barcelona with a strong effect of the saltier and cooler Northern current flowing southwest. The Central zone, includes the wide platform off the coast of Tarragona down to the Gulf of Valencia, characterized by the inshore influence of the Northern current while offshore is influenced by a countercurrent flowing northeastward, the Balearic Current, which creates a number of mesoscale circulation structures (eddies). Finally, the Southern zone, includes the Ibiza Channel and the coast off Alicante down to Cartagena (Fig 1), is mostly influenced by less salty Atlantic water recently entering the Mediterranean through the Strait of Gibraltar and flowing northeastward. Parametric and non-parametric tests (t-tests, Kruskal-Wallis and Wilcoxon-Mann-Whitney pairwise tests) were performed by zone with larval abundance and diversity data (homogeneity of the variance was checked through Levene test and normality of the residuals through Shapiro-Wilk test). A permutational multivariate analysis of variance using Jaccard distance matrix was performed to check for significant differences in the composition of the communities of the three zones, using the Vegan package [46] in R [47].

For community analysis, we assessed the linkage between the environmental variables (SST, SSS, SSF) and larvae abundances by combining non-metric multi-dimensional scaling analysis (nMDS) with generalized additive models (GAMs). First, the nMDS was fitted after scaling the data to represent the species ordination onto a 2D space. We used the metric version of the

Bray-Curtis dissimilarity index, called Jaccard index, which yielded a suitable representation of the species ordination in the two dimensions (stress plot of $r^2 = 0.99$). A stable solution was found using several random starts, using the Vegan package [46] in R [47]. Bray–Curtis and Jaccard indices are rank-order similar, and some other indices become identical or rank-order similar after some standardizations, especially with presence/absence transformation of equalizing site totals with decostand. Jaccard index is metric, and probably should be preferred instead of the default Bray-Curtis which is semimetric. Then, the association between the species ordination and the environmental variables was achieved using GAMs [48], following the Vegan package. The GAMs allowed to create smooth surfaces over the species ordination to identify species links to specific environmental conditions.

## Results

A total of 20,022 decapod crustacean larvae belonging to 18 families (plus the three infraorders kept separate, Anomura, Achelata and Brachyura) were identified from plankton samples. No larvae of Astacidea or Polychelida were found. Higher values of estimated zooplankton volume were found in the Northern zone and the Ebre Delta (Fig 2), and this variable was significantly correlated to total decapod larval abundance (p < 0.01, $r^2 = 0.35$).

The average density values for all decapod larvae combined were comprised between 1.53 and 9464.47 individuals per 1000 m$^3$. The most abundant general taxonomic groups were brachyuran crabs and penaeoid shrimps (Table 1). The former represented more than 10% of all

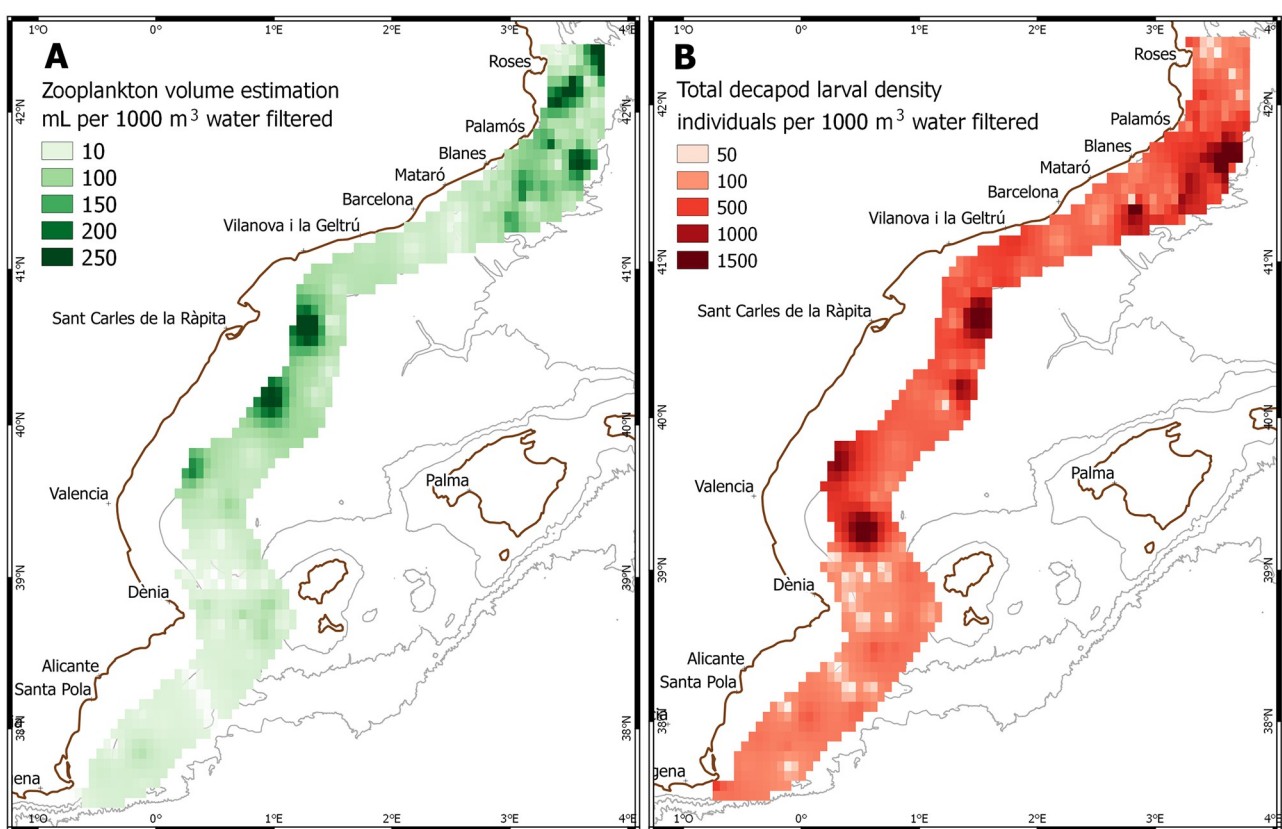

**Fig 2.** A: Zooplankton volume estimation in samples (mL per 1000 m$^3$ water filtered); B: Total decapod larval density (individuals per 1000 m$^3$ water filtered).

**Table 1. Total number (N), average density of individuals per 1000 m³ with standard deviation and frequency (F) of decapod crustacean larvae caught with neuston sledge.** NID: Non identified.

| Taxon | N | Average density (ind. 1000 m⁻³) | F (%) |
|---|---|---|---|
| **Suborder Dendrobranchiata** | | | |
| Superfamily Penaeoidea | 6839 | 146.07 ± 929.18 | 63.37 |
| Superfamily Sergestoidea | 2610 | 50.57 ± 75.27 | 72.28 |
| **Suborder Pleocyemata** | | | |
| Infraorder Caridea | 1021 | 20.04 ± 48.13 | 47.52 |
| Infraorder Axiidea | 6 | 0.12 ± 0.54 | 4.95 |
| Infraorder Gebiidea | 53 | 1.05 ± 5.02 | 9.90 |
| Infraorder Stenopodidea | 3 | 0.06 ± 0.46 | 1.98 |
| Infraorder Anomura | 2075 | 44.46 ± 144.17 | 39.60 |
| Infraorder Achelata | 55 | 1.12 ± 4.20 | 14.85 |
| Infraorder Brachyura | 7194 | 147.00 ± 405.95 | 49.50 |
| NID | 100 | 2.04 ± 7.57 | 23.76 |
| **TOTAL** | **20,022** | **217.90 ± 945.26** | |

decapod larvae in all three zones studied, while penaeoid shrimps were clearly dominant in the Northern zone and practically absent in the Southern zone, in favour of sergestoid shrimps (Fig 3).

The Shannon diversity index showed significantly higher values in the Central zone than in the Northern zone (Fig 4; pairwise t-test, 85 degrees of freedom, t = 3.35, p = 0.017). The species richness was also significantly higher in the Central zone than in the Northern and Southern zones (Fig 4; Wilcoxon rank sum test, W = 2016 and 990, respectively; p = 0.001 and p = 0.025, respectively). Total larval density values were significantly lower in the Southern zone than in the Central zone (Fig 4; Wilcoxon rank sum test, W = 590, p = 0.02). The differences in zooplankton estimated volume were significant (Fig 4) but the paired tests could not be run due to missing data. No significant differences were found in evenness index values. The permutational analysis of variance revealed significant differences in the composition of the Central and Northern communities (F = 2.94, p = 0.018).

As for shrimp larvae, the most abundant taxa were the deep-sea blue and red shrimp *Aristeus antennatus* and the small mesopelagic shrimps of the family Sergestidae (Table 2). These taxa were also the most frequent, representing more than 50% of all decapod larvae on average. Overall, 89% of all Dendrobranchiata larvae identified were in the protozoea phase. The three Dendrobranchiata larvae analyzed with molecular methods were identified as *Penaeus kerathurus* (N = 2, mysis III and IV stages) and *Sicyonia carinata* (N = 1, mysis I stage).

For clarity, we will present further results on 12 taxa that showed either the highest average density values, the highest frequency values, or are sold commercially (some taxa fell into the three categories at once): *Aristeus antennatus*, *Penaeus kerathurus*, *Parapenaeus longirostris*, *Sycionia carinata*, Sergestidae, *Alpheus glaber*, *Athanas nitescens*, *Lysmata* spp., *Processa* spp., *Plesionika* spp., *Upogebia* spp., and Achelata. Regarding non-commercial species (Fig 5), the family Sergestidae is followed in abundance values by the snapper shrimps *A. glaber* and *A. nitescens* (Fig 5C and Fig 5D), the cleaner shrimp genus *Lysmata* (Fig 5E), the small epibenthic shrimps of the genus *Processa* (Fig 5F), the mud shrimp *Upogebia* (Fig 5G), and the Mediterranean rock shrimp *Sicyonia carinata* (Fig 5A). The Sergestidae were the only taxa of which larvae were present in the Southern zone, below the Ibiza Channel, where the occurrences of other taxa were marginal. For commercial species (Fig 6), our results showed a scarce presence

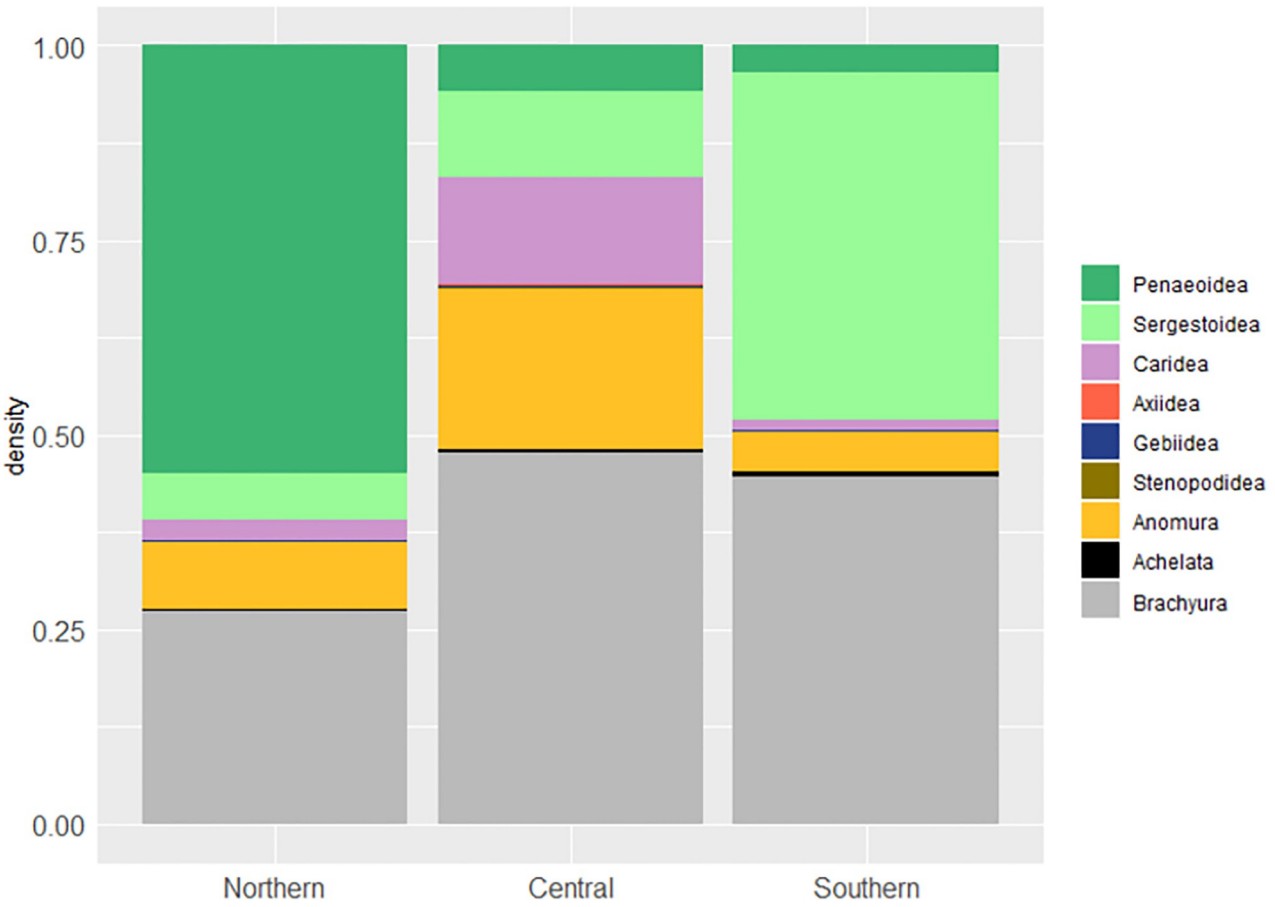

**Fig 3. Average relative larval density (in individuals per 1000 m³) of taxa in the 3 zones studied.**

of the larvae of *P. longirostris*, as well as slightly higher abundance values for *P. kerathurus*, the genus *Plesionika* and the lobsters of the infraorder Achelata.

For spatial representation of the summer decapod larval community, only the selected taxa that were identified to species level were considered, i.e. *A. antennatus*, *A. glaber*, *P. kerathurus*, *A. nitescens*, *S. carinata* and *P. longirostris*.

The species ordination was significantly linked to sea surface temperature, salinity and fluorescence (p = 0.049, p = 0.001 and p = 0.003, respectively; Fig 7). *A. antennatus* larvae were associated with colder (Fig 7A) and more saline (Fig 7B) waters around 24 ºC and 38 in the Northern zone, while *P. longirostris*, *S. carinata*, and *A. nitescens* were mainly associated with low-salinity (Fig 7B) and low-fluorescence (Fig 7C) waters in the Central and Southern zones (37.8 and 0.046 μg/L).

## Discussion

In this mesoscale analysis, we have shown the composition and diversity patterns of the summer decapod crustacean larval community in the GSA6 area of the Mediterranean Sea. Zooplankton volume values were higher in the Northern zone, cut by deep submarine canyons that have been reported as biodiversity hotspots [49, 50]. However, our findings of higher values of diversity and species richness in the Central zone, between the Ebro Delta and the Ibiza

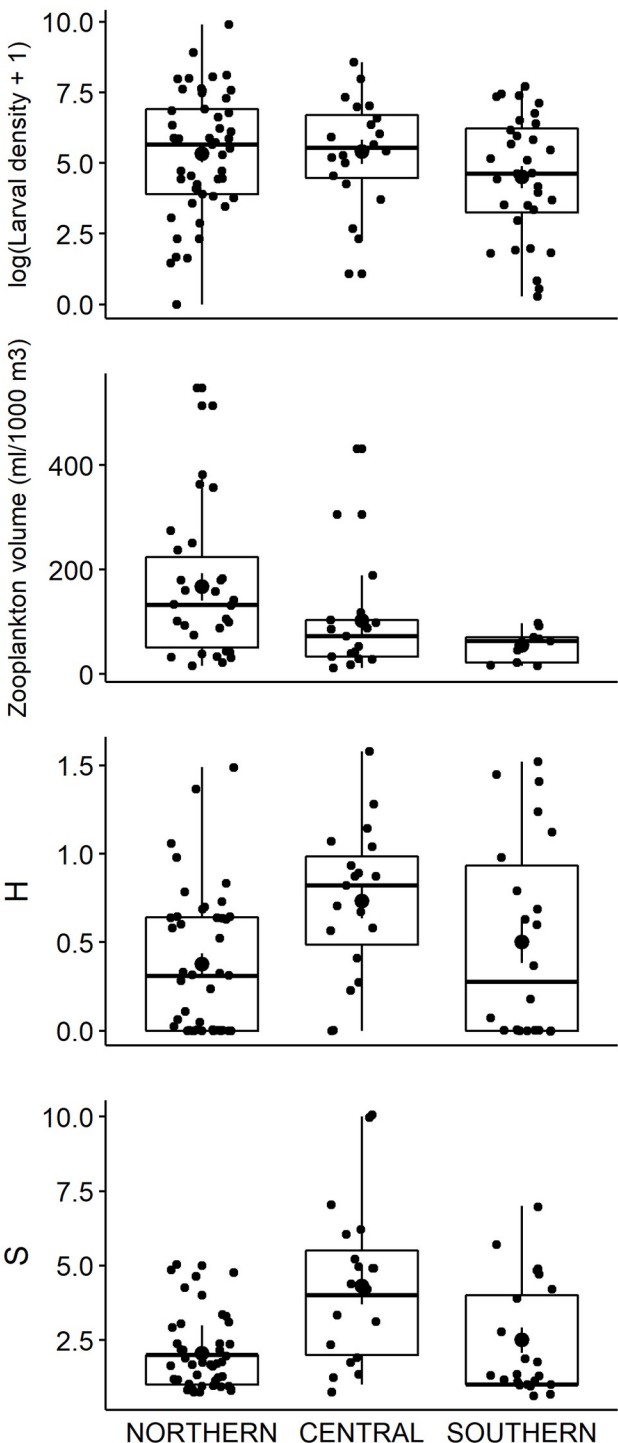

**Fig 4. First quartile, median, third quartile and standard error for total decapod larval density (log (individuals per 1000 m³ + 1)), Zooplankton estimated volume (mL per 1000 m³), Shannon diversity index (H) and species richness (S) for the three zones of GSA 6.**

**Table 2. Total number (N), average density of individuals per 1000 m³ with standard deviation, and frequency in relation to all decapod larvae (F) of shrimp larvae caught with neuston sledge.** NID: Non identified.

| Taxon | N | Average density (ind. 1000 m⁻³) | F (%) |
|---|---|---|---|
| **Suborder Dendrobranchiata** | | | |
| Superfamily Penaeoidea | | | |
| Family Aristeidae | | | |
| *Aristeus antennatus* | 6571 | $140.82 \pm 928.74$ | 56.44 |
| Family Benthesicymidae | | | |
| *Gennadas elegans* | 10 | $0.26 \pm 2.03$ | 1.98 |
| Family Penaeidae | | | |
| *Parapenaeus longirostris* | 12 | $0.25 \pm 2.16$ | 2.97 |
| *Penaeus kerathurus* | 209 | $3.93 \pm 12.39$ | 17.82 |
| Family Sicyonidae | | | |
| *Sicyonia carinata* | 34 | $0.75 \pm 3.40$ | 8.91 |
| Family Solenoceridae | | | |
| *Solenocera membranacea* | 3 | $0.06 \pm 0.33$ | 2.97 |
| Penaeoidea NID | 2 | $0.04 \pm 0.43$ | 0.99 |
| Superfamily Sergestoidea | | | |
| Family Sergestidae | | | |
| *Allosergestes sargassi* | 1 | $0.02 \pm 0.21$ | 0.99 |
| *Deosergestes corniculum* | 23 | $0.47 \pm 1.83$ | 8.91 |
| *Deosergestes henseni* | 53 | $1.12 \pm 2.89$ | 17.82 |
| *Eusergestes arcticus* | 123 | $2.02 \pm 9.84$ | 12.87 |
| *Sergestes atlanticus* | 30 | $0.65 \pm 3.79$ | 7.92 |
| *Robustosergia robusta* | 6 | $0.09 \pm 0.63$ | 2.97 |
| *Parasergestes vigilax* | 92 | $2.07 \pm 12.44$ | 4.95 |
| Sergestidae NID | 2282 | $44.14 \pm 72.98$ | 67.33 |
| Dendrobranchiata NID | 5 | $0.09 \pm 0.57$ | 2.97 |
| **Suborder Pleocyemata** | | | |
| Infraorder Caridea | | | |
| Family Acanthephyridae | | | |
| *Acanthephyra* spp. | 17 | $0.31 \pm 1.01$ | 10.89 |
| Family Palaemonidae | | | |
| *Palaemon macrodactylus* | 11 | $0.21 \pm 1.03$ | 5.94 |
| *Palaemon serratus* | 7 | $0.12 \pm 0.82$ | 3.96 |
| *Palaemon* sp1 | 6 | $0.12 \pm 0.76$ | 2.97 |
| *Palaemon* sp2 | 2 | $0.04 \pm 0.39$ | 0.99 |
| Palaemonidae NID | 24 | $0.71 \pm 6.88$ | 1.98 |
| Family Pontoninae | | | |
| *Pontonia* spp. | 8 | $0.16 \pm 1.14$ | 1.98 |
| Pontoninae NID | 6 | $0.12 \pm 0.85$ | 1.98 |
| Family Alpheidae | | | |
| *Alpheus glaber* | 375 | $6.89 \pm 20.59$ | 28.71 |
| *Athanas nitescens* | 67 | $1.17 \pm 5.51$ | 10.89 |
| Family Thoridae | | | |
| *Eualus cranchii* | 1 | $0.02 \pm 0.17$ | 0.99 |
| *Eualus occultus* | 24 | $0.49 \pm 2.37$ | 9.90 |
| Family Lysmatidae | | | |
| *Lysmata* spp. | 96 | $2.08 \pm 7.28$ | 21.78 |

*(Continued)*

**Table 2.** (Continued)

| Taxon | N | Average density (ind. 1000 m$^{-3}$) | F (%) |
|---|---|---|---|
| Family Processidae | | | |
| *Processa* spp. | 87 | 1.60 ± 5.67 | 15.84 |
| *Processa modica caroli* | 4 | 0.07 ± 0.49 | 1.98 |
| Family Pandalidae | | | |
| *Plesionika* spp. | 236 | 4.55 ± 14.61 | 30.69 |
| *Pandalina* spp. | 15 | 0.30 ± 2.13 | 1.98 |
| Pandalidae sp1 | 1 | 0.02 ± 0.15 | 0.99 |
| Pandalidae NID | 31 | 0.53 ± 5.18 | 1.98 |
| Family Crangonidae | | | |
| *Aegaeon* spp. | 2 | 0.05 ± 0.48 | 0.99 |
| *Aegaeon lacazei* | 1 | 0.02 ± 0.25 | 0.99 |
| *Philocheras* spp. | 2 | 0.04 ± 0.30 | 1.98 |
| Caridea NID | 33 | 0.70 ± 2.95 | 8.91 |
| Infraorder Axiidea | | | |
| Family Callianassidae | | | |
| *Callianassa subterranea* | 3 | 0.05 ± 0.39 | 1.98 |
| CALSL16* | 1 | 0.02 ± 0.17 | 0.99 |
| Calianassidae NID | 2 | 0.05 ± 0.34 | 1.98 |
| Infraorder Gebiidea | | | |
| Family Upogebiidae | | | |
| *Upogebia* spp. | 53 | 1.05 ± 5.02 | 9.90 |
| Infraorder Stenopodidea | | | |
| Family Stenopodidae | | | |
| *Stenopus spinosus* | 3 | 0.06 ± 0.46 | 1.98 |

*Dos Santos and González-Gordillo (2001)

Channel, may be related to the influence of the Ebro river discharge, which would be more apparent in the surface layer. The high variability of the data, especially in the Southern and Central zones, prevent us from drawing more robust conclusions.

In a study of the eastern Spanish Mediterranean coast, Pastor-Prieto et al. [45] found a clear contrast in surface water characteristics between the Southern zone, with new Atlantic water that enters the Mediterranean basin through the Strait of Gibraltar, and the Central and Northern zones, with old Atlantic water carried by the Northern Current. In our samples, the zooplankton volume values were clearly higher in the northern and central zones, associated with lower-salinity new Atlantic water, while the general decapod larval density was found to be uniform throughout the study area.

In line with a smaller-scale study in the area of the submarine canyon off Blanes, our results show the larvae of the deep-sea blue and red shrimp *A. antennatus* as one of the dominant species in this period of the year [20]. They were associated to cold, high-salinity waters which would correspond to the old Atlantic water of the Northern zone.

Larvae of the meso- and epipelagic shrimp of the family Sergestidae are distributed throughout the area in the summer. In the eastern Spanish Mediterranean coast, this family is represented by seven species from the former genera *Sergestes* and *Sergia* whose reproductive period has not yet been studied thoroughly, and previous community studies in the

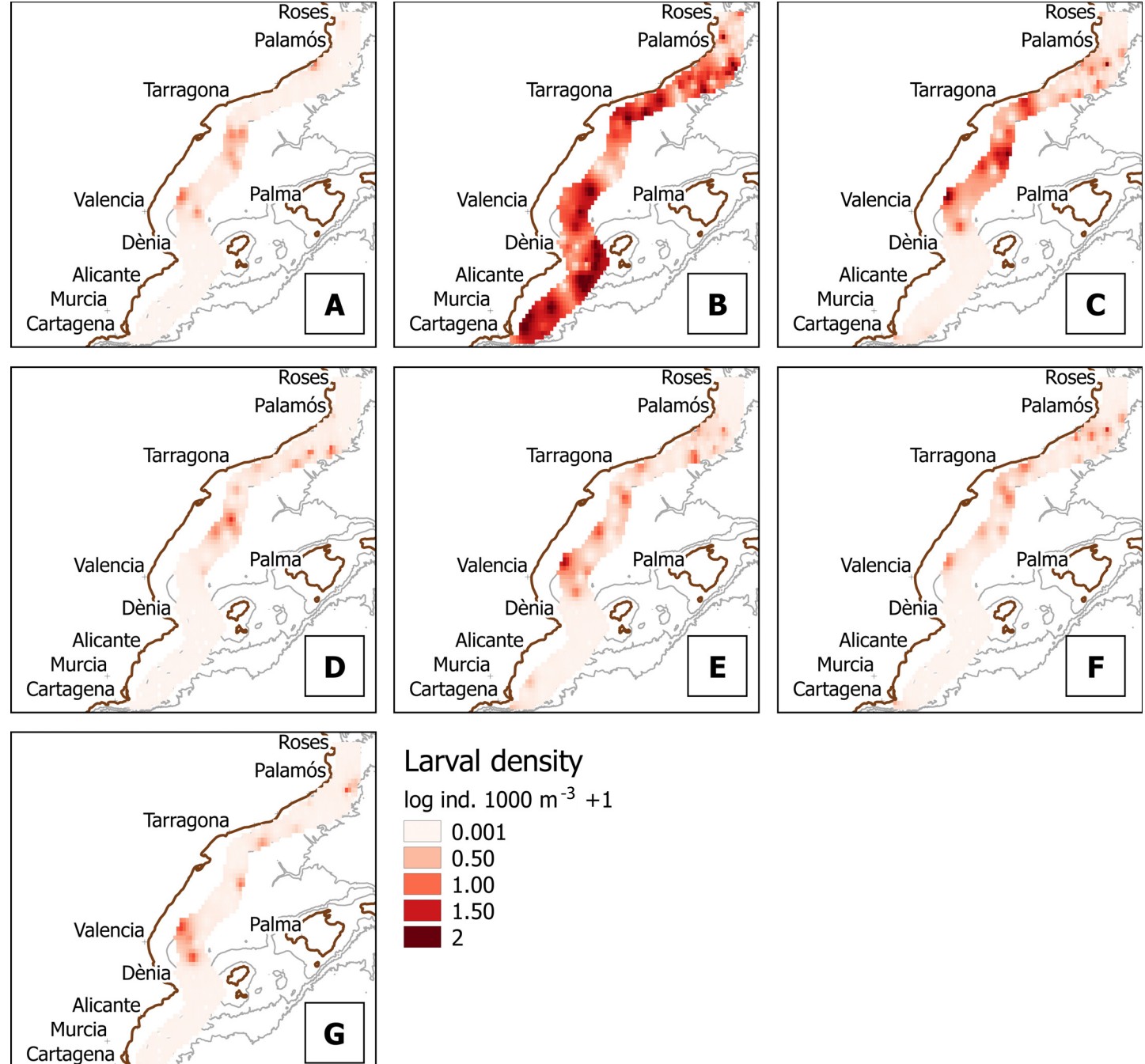

**Fig 5. Larval density in log10 (individuals 1000 m⁻³ + 1) for the most abundant and/or frequent taxa of no commercial interest.** A: *Sycionia carinata*, B: Sergestidae; C: *Alpheus glaber*; D: *Athanas nitescens*; E: *Lysmata* spp.; F: *Processa* spp.; and G: *Upogebia* spp.

Northwestern Mediterranean Sea either report their larvae to occur only in the winter [16] or to be present throughout the year and more abundant in the summer [19].

Regarding the rest of relevant taxa in the studied communities, the relative importance of snapper, cleaner and mud shrimps along with the genus *Processa* agrees with prior decapod larval studies in the area [16, 19]. The genus *Plesionika* is widely represented in the area

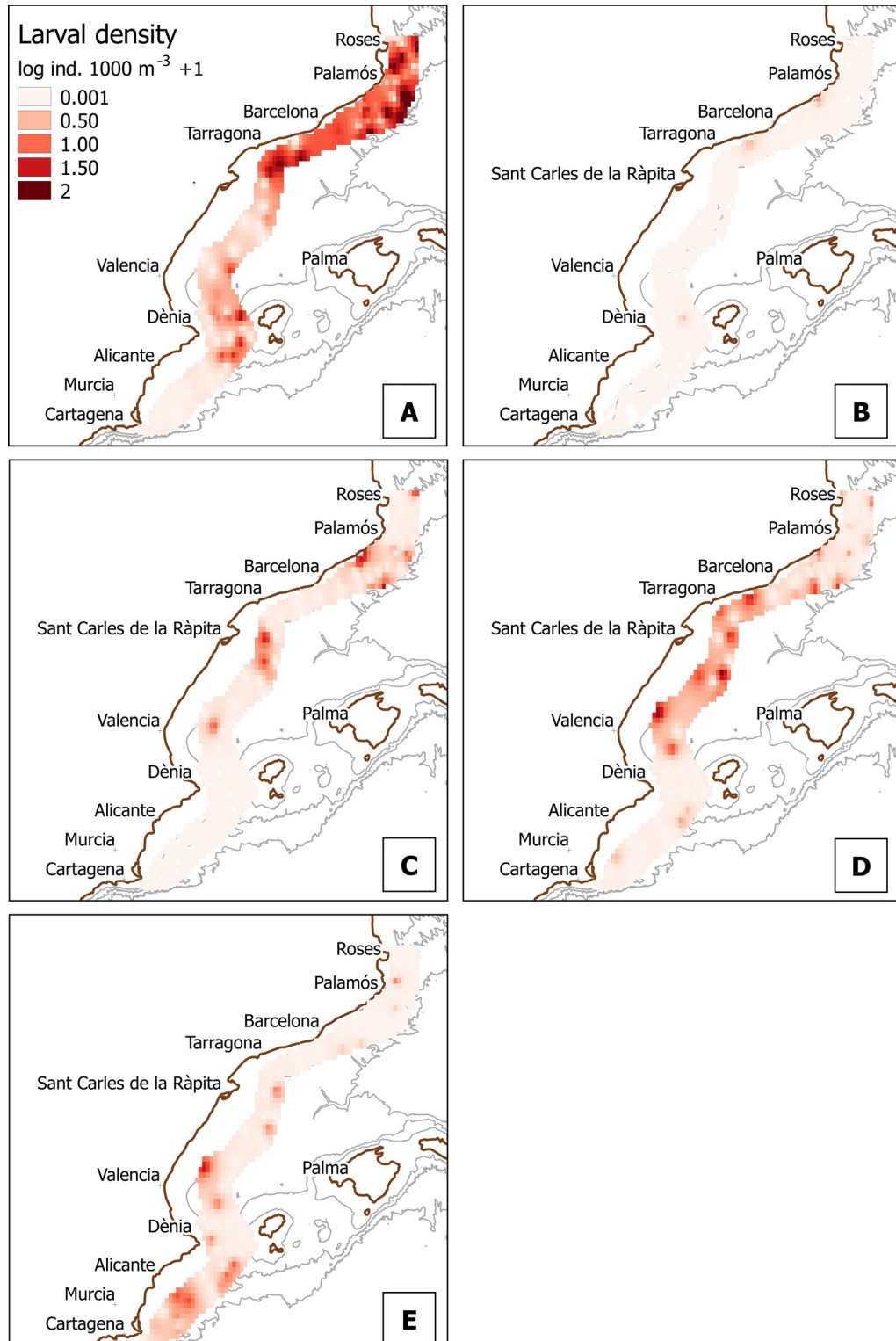

**Fig 6. Larval density in log (individuals 1000 m$^{-3}$ + 1) for taxa of commercial interest.** A: *Aristeus antennatus*; B: *Parapenaeus longirostris*; C: *Penaeus kerathurus*; D: *Plesionika* spp.; and E: lobsters (Achelata).

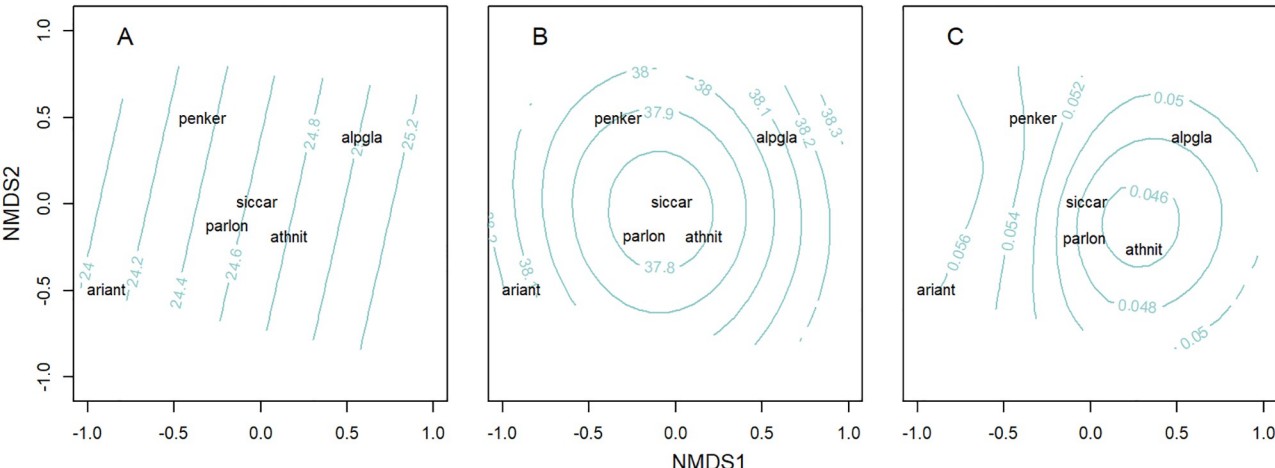

**Fig 7.** NMDS species ordination over contours, representing environmental gradients for A: sea surface temperature (°C), B: sea surface salinity (PSU), C: sea surface fluorescence (µg/L). ariant: *Aristeus antennatus*; parlon: *Parapenaeus longirostris*; penker: *Penaeus kerathurus*; siccar: *Sicyonia carinata*; alpgla: *Alpheus glaber*; athnit: *Athanas nitescens*.

through 5 species (*P. heterocarpus*, *P. edwardsii*, *P. giglioli*, *P. martia* and *P. acanthonotus*) that vary in the timing of their reproductive period, although they generally spawn in spring or early summer [4]. This diversity can explain their relatively abundant and uniform presence throughout the study area. On the other hand, the larvae of the common deep-water rose shrimp *P. longirostris* have been reportedly difficult to find in plankton surveys [19, 24, 51, 52], and they were also scarcely found in the present study, since the species spawns throughout the year, but the spawning peaks occur in late spring and early autumn [53, 54]. To our knowledge, this study still constitutes the third finding of these larvae in the plankton, and the highest number of them ever caught in the Mediterranean Sea [18, 19].

The presence of larvae of the caramote prawn (*P. kerathurus*) off Roses (the northernmost part of the study area) is in line with fisheries studies that report this species in the catches [55]. However, it is worth noting that these larvae have also been found a little further south, around Blanes submarine canyon (46 individuals per 1000 m³), where this species is not regularly fished, whereas they are only faintly present in the samples off the Ebro Delta, where its most relevant fishing grounds are located. The larvae of *P. kerathurus* have been previously caught in nearby areas, off the coast of Barcelona, far away from fishing grounds or habitat of the species, although they were most abundant in February and absent during the summer [16]. The same author later reported these larvae being absent from the Ebro Delta [17]. Although the identification of some of these larvae in the mysis stage was in fact confirmed by molecular methods in this study, we cannot rule out the possibility that the first protozoeal stage, which represents half of the larvae we found for this species, is not easily differentiated from that of *Penaeus japonicus*, given the lack of recent information [56, 57]. In addition, the absence of *P. kerathurus* larvae we observed near the Ebro delta may be due to our sampling stations being located too far from the coast, since the most common habitat depths for this species range from 50 to 300 m depth [58].

The total absence of *Pasiphaea* larvae is also remarkable, since it is reportedly one of the most abundant decapod crustacean genera in the area [59] and, in the case of the two species found in the studied area, i.e. *P. multidentata* and *P. sivado*, their peak reproductive period is in the summer [4]. However, they were only reported in very low numbers in a previous study in the Balearic Sea [19] and absent from other studies in the area [16, 17].

As for Achelata larvae, advances are still necessary in the identification of these phyllosomae to species level, although we can presume our larvae to belong to the Scyllarid family, and more probably to *Scyllarus pygmaeus*, which is the most common Scyllarid larva found in the Mediterranean Sea [60].

Brachyuran megalopae were found consistently in the neuston, as reported by Abelló and Guerao [61], although brachyuran zoeae were more abundant in our samples (average density of 20.15 versus 410.10 ind. 1000 m$^{-3}$, respectively). This could be due to the distance of the sampling sites to the coast, since megalopae are associated with less saline, inshore waters [61].

The mesoscale approach of this study allows for a broad view of the summer decapod larval community of the Eastern Spanish Mediterranean coast. Our results have given information on species that might not be abundant in the plankton and therefore absent in a smaller-scale sampling, but are ecologically or commercially relevant nonetheless. These results set a stepping stone for the development of larval ecology studies of these species.

## Supporting information

**S1 Table. Information of sampled stations.** Lat.: Latitude; Long.: Longitude. Depth corresponds to bottom depth.
(DOCX)

## Acknowledgments

The authors would like to thank all the participants in the CONECTA cruise on board R/V *García del Cid*, and especially Drs. Pilar Olivar and Ana Sabatés (ICM-CSIC) for their assistance in plankton sampling methodology. Finally, we wish to thank Dr. Antonina Dos Santos (IPMA) for the dedicated training in decapod larvae identification.

## Author Contributions

**Conceptualization:** Marta Carreton, Guiomar Rotllant, Diego Castejón, Nixon Bahamón, Joan B. Company.

**Data curation:** Marta Carreton, Guiomar Rotllant, Nixon Bahamón, Joan B. Company.

**Formal analysis:** Marta Carreton, Diego Castejón, Nixon Bahamón.

**Funding acquisition:** Guiomar Rotllant, Joan B. Company.

**Investigation:** Marta Carreton, Guiomar Rotllant, Diego Castejón.

**Methodology:** Marta Carreton, Guiomar Rotllant, Diego Castejón, Nixon Bahamón, Joan B. Company.

**Project administration:** Guiomar Rotllant, Joan B. Company.

**Resources:** Joan B. Company.

**Supervision:** Guiomar Rotllant, Nixon Bahamón, Joan B. Company.

**Validation:** Marta Carreton, Guiomar Rotllant, Nixon Bahamón, Joan B. Company.

**Visualization:** Marta Carreton, Diego Castejón, Nixon Bahamón.

**Writing – original draft:** Marta Carreton.

**Writing – review & editing:** Guiomar Rotllant, Diego Castejón, Nixon Bahamón, Joan B. Company.

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
