## [Decision Letter · Decision Letter 0]

23 Jun 2022

PONE-D-22-14466Decapod crustacean larval communities along the eastern Spanish Mediterranean coastPLOS ONE

Dear Dr. CLARET,

Thank you for submitting your manuscript to PLOS ONE. After careful consideration, we feel that it has merit but does not fully meet PLOS ONE’s publication criteria as it currently stands. Therefore, we invite you to submit a revised version of the manuscript that addresses the points raised during the review process.

The reviewers feel this paper presents interesting data resulting from an intense sampling and classification effort, and deserve publication after minor revisions. The title is considered to be a bit showy and not exactly fitting the paper contents. Some methodological aspects need further clarification.

We look forward to receiving your revised manuscript.

Kind regards,

Antonio Medina Guerrero, Ph.D.

Academic Editor

PLOS ONE

Journal Requirements:

"This research was carried out under project CONECTA (CTM2014-54648-C2) funded by the Spanish Miniterio de Economía y Competitividad. MC benefited from a FPU2015 grant from the Spanish Ministerio de Educación. The funders had no role in study design, data collection and analysis, decision to publish, or preparation of the manuscript. "

Reviewers' comments:

Reviewer's Responses to Questions

**Comments to the Author**

1. Is the manuscript technically sound, and do the data support the conclusions?

Reviewer #1: Yes

Reviewer #2: Partly

2. Has the statistical analysis been performed appropriately and rigorously? 

Reviewer #1: Yes

Reviewer #2: Yes

3. Have the authors made all data underlying the findings in their manuscript fully available?

Reviewer #1: Yes

Reviewer #2: No

4. Is the manuscript presented in an intelligible fashion and written in standard English?

Reviewer #1: Yes

Reviewer #2: Yes

5. Review Comments to the Author

Reviewer #1: Dear authors,

I find this paper interesting, not pretentious, well written and being the result of a large amount of work. The classification of marine larval organisms is arduous and capable people are lacking. I think this work merits publication in this journal after minor revision. Please find my comments below.

Line 71. The meaning of GSAs is known to Mediterranean scientists, but other readers may not be familiar with this GFCM divisions

Line 73. I would refer here to Figure 1.

Line 76. There are some more works in the W Med, e.g. Pires et al.,2018. https://doi.org/10.12681/mms.2006

Line 87. I would move this table to the supplementary material, refer to Fig.1. Table 1 occupies a lot of space and the level of detail offered (Lat and Longs for example) are not needed "inmediately"

Line 127-131. Besides a-priori analyses, it is always interesting to conduct a data-driven analysis (e.g. clustering/ordination methods ), without considering the three zones. in this way, you can extract spatial patterns that may not be related to your pre-conceptions. Note: I see that you later conduct NMDS..

Methods: Despite GAMS are widely used, in my view GLMs should be preferred (can use linear models that allow for curved responses) if you do not have a clear view of why your data may be non-linear in a particular way (specially if you have few data in some areas of your graph). In general, I miss the results (values) of the GAMS/NMDS analyses and a better explanation of those results. How good was this approach?

Line 175. In general, in all tests I'd include the statistical test used, its value, etc., unless not required by the journal

Line 178. These p values are very marginal, If the analysis was bayesian, probably you would not claim differences to be too credible. I'd be cautious with these interpretations. Later I saw you include one sentence on this in the Discussion, but still, you expand on these results.

Line 190. On average

Line 214. Log10 or Ln? You mention Log10 in the Methods

Line 226. Again, p values tell not too much, including sample size, degrees of freedom and the test value is recommended.

Line 226-229. I wonder if you are no missing some key predictive variables, such as sediment type where some adults dwell

Line 251 "although followed...." that sentence is a bit difficult to understand

Line 255. Do the samples from that study come from a different or the same survey?

Reviewer #2: This work is the result of two intense efforts, the first of them in collecting a huge number of nektonic samples (at 101 stations) along the Iberian Mediterranean coast, and the second one in identifying 20,022 decapod larvae. This is an impressive task! Congratulations to the authors.

I think that is a correct work respect to methods, treatment and analysis of data, and discussion of results.

Anyway, I have some questions that I think that must be addressed by the authors before this manuscript be acceptable for publication.

1)The first question is related to the title and some sentences along the manuscript that from my point of view do not correspond with the real content of the work. I mean, as authors say at some point (lines 15-16), they “present the summer mesoscale larval distribution of several species of commercial interest”, as well as the objective of the work (lines 79-81): “to describe the summer community…”. But the title does not reflect this exactly, and it could create false expectations in the readers like myself. Therefore, I suggest adjusting the title to the actual content.

2) The second question refer to the methods for collecting. Authors used a neustonic sledge, with a flowmeter to calculate the volume of filtered water. But I miss some important data about the moment of the day for collection because we know that nekton composition is not the same during the day or in the night. There are significant variations in species composition and abundances. Therefore, I think that this is an important information to understand the results. As well as the period of time that the neustonic sledge was towed. I know that finally with the flowmeter you can compare samples taken in different periods of time (from few minutes to longer), but with longer periods the probability of collecting species with lower densities increase. Therefore, to help to understand the results for readers I suggest including this information in the Method section.

3) Respect to the area of study, authors selected three zones: Northern, Central, and Southern; but I cannot see the criterion for the limits of each one. According to figure 1, the limit between “Southern” and “Central” is not exactly the Ibiza Channel, and the limit between “Central” and “Northern” is not between Blanes and Palamos, why? Authors mention these two barriers in the Introduction (lines 66-78), so why they did not use this information to define the three zones? Instead, they define 3 zones with a clear different number of stations (Southern: 31, Central: 20, Northern: 50) and different distribution (density of stations at Northern zone clearly the highest). In the Northern zone (figure 1) is easy to see distribution of several sampling stations forming perpendicular lines to the coast line (radially), but not in the other two zones. I suggest to the authors to describe the criterion to stablish the 3 zones, and the distribution and density of stations, and how this could affect (or not) to the results and the analysis.

4) I know that is impossible to take samples at 101 stations along hundreds of miles simultaneity, but between the first samples and the last there are more than a month. This affects the results in two ways, different reproductive periods or pike of larval emission depending on the species could be not take into account, and these samples taken in this period do not represent “the “summer” distribution of…”. Summer start one month before starting the sampling and finish a month later. I know that authors mention this in the Discussion section (lines 258-312). But, for this reason some sentences must be corrected considering this. For example: lines 236-237, and 313-314,

5) I am not sure whether this is a rule of the journal, but in any case, I do not understand why all scientific names of species in the references are not in italics (see lines 338, 345, 346, 351, 354, 369, 378, 380, etc…)? The journal names are in italics, so is not a problem with this letter format.

Other minor comments and corrections as follow:

Line 10. Authors mention the taxa identified at the lowest level (shrimp, sensu lato), and those left at infraorder, but I miss two infraorder well-represented in marine waters, Astacidea and Polychelida, and my question is, are not mentioned because are not considered or because no larvae of these taxa were found? Absence of species is also an interesting information. For example, an Astacidea, like Nephrops norvegicus is also an commercial important species, and their larvae are extremely difficult to find in plankton samples.

Lines 26-27. Probably this first sentence must be supported by a reference.

Lines 50-51. “the eggs hatch directly into the first zoeal stage”. This is not right. There are exceptions, species with abbreviated development, or direct development. In these cases they can hatch as decapodite/megalopa (not zoea) or even as first crab/juvenile. Correct accordingly.

Lines 53-65. In this paragraph I suggest citing works like:

Brandao, M. C., Freire, A. S. & Burton, R. S. (2016) Estimating diversity of crabs (Decapoda: Brachyura) in a no-take marine protected area of the SW Atlantic coast through DNA barcoding of larvae. Syst. Biodivers. 14(3), 288–302.

Marco-Herrero, E., Cuesta, J.A. & González-Gordillo, J.I. (2021) DNA barcoding allows identification of undescribed crab megalopas from the open sea. Scientific Reports 11:1.

To show the use of molecular techniques in larval studies.

Lines 66-78.

In this paragraph I suggest to cting this work, if after read it you consider that allow to explain the barriers in the studied area:

Ojeda, V., Serra, B., Lagares, C. et al. (2022) Interannual fluctuations in connectivity among crab populations (Liocarcinus depurator) along the Atlantic-Mediterranean transition. Scientific Reports, 12, 9797.

Line 71. I suggest that first time cited in the text you mention it in full, for people that unknow the meaning of “GSA 6”.

Line 73. “Carreton et al., in press”, I did not find this citation in references. I understand that papers in press can be cited and included in the reference list.

Lines 66-78. Considering all the information of this paragraph, I suggest including these barriers in the figures 1, 2, 5 and 6. (see for example figure 1 in Ojeda et al. (2022))

Lines 104-106. Here, and in the paragraphs 191-193 and 289-294, authors explain that used molecular identification for three larvae. I miss a little bite of more information. What genes were used? COI? 16S? other? Authors say “following the method detailed by Carreton et al. (2019)”, in that work they used different primers, for different species, so I am not suggesting to include here all that information about primers and pcr protocols, but at least the gene or genes used, and the accession codes to Genbank (or BOLD, if COI) of the sequences obtained.

Lines 154-156. As commented above, I suggest mentioning the remaining Infraorders (as Astacidea and Polychelida), and comment that larvae of these taxa where not found in your samples. The absence, or the negative result, is also an interesting data.

Lines 156-157. In fig. 2A I can see high volume of zooplankton in the Central zone too, especially in the Ebro Delta area.

Lines 204-207. In this analysis of non-commercial species, there are a comparison of Sergestidae (higher abundance) with the rest of taxa, but in the order of more to less represented authors forget to mention S. carinata (Fig. 5A). In this comparison, they use species (S. carinata, Alpheus glaber, Athanas nitescens), genus (Lysmata, Processa, Upogebia) and family level (Sergestidae), but in Table 3, we can see the level of identification of these taxa in families too. So, maybe could be all compared at this level? In some cases, for example S. carinata, is the same, but in other as Alpheidae would be the sum of Alpheus glaber and Athanas nitescens. It is just a suggestion.

Lines 236-237. Insert in this sentence the temporal window: “summer”.

Line 352. Include the name of the journal (PLOS ONE, I think), and volume.

Line 387. Complete this reference with number of Leaflet and number of pages.

Lines 401-406. I unknow the rules for references of this journal, but I do not understand the order of the three papers by Fuste. First 1987, second 1989 and third 1982.

Line 404. Complete this reference with number of volume and number of pages.

Line 406. Complete this reference with number of volume and number of pages.

Lines 432-434. This is a PhD. thesis? I suggest completing the information, University? Number of pages? Published (printed or online), unpublished?

Lines 442-444. This is also a PhD. Thesis? I suggest the same that above.

Lines 455-456. This reference is confused. This is a chapter of book, so I miss data about editorial, country, pages, etc., and what about (Anger, 2001)?? Please, correct accordingly.

Lines 480-483. Like above, this is a book chapter? I miss data.

Lines 498-500. I think that the name of the journal must be included, and what mean (Lc)?

6. PLOS authors have the option to publish the peer review history of their article (what does this mean?). If published, this will include your full peer review and any attached files.

Reviewer #1: No

Reviewer #2: No

---

## [Author Response · Author response to Decision Letter 0]

9 Sep 2022

Dear Editorial Team,

The authors would like to thank the reviewers for their insightful comments, which we hope to have adressed as thoroughly as intended. Also, the fact that they recognized our sampling and identification effort spurred us to improve the manuscript to achieve the quality required for publication. 

The images have now been modified using the PACE tool. The references list has been revised according to the reviewers’ comments and updated to meet the journal’s standards. All funding-related text has been removed from the manuscript, and we kindly ask you to modify the funding section of our online submission on our behalf to state the following: 

“This research was carried out under project CONECTA (CTM2014-54648-C2) funded by the Spanish Miniterio de Economía y Competitividad. MC benefited from a FPU2015 grant from the Spanish Ministerio de Educación. The funders had no role in study design, data collection and analysis, decision to publish, or preparation of the manuscript.”

We here enclose our detailed response to the reviewers’ comments and suggestions. All line numbers refer to the Track Changes version of the revised manuscript.

Sincerely,

Joan B. Company

Corresponding author

 

Reviewer #1: 

1) Line 71. The meaning of GSAs is known to Mediterranean scientists, but other readers may not be familiar with this GFCM divisions.

The sentence now reads: “... in this area, designed as the fisheries management unit Geographical Subarea 6 (or GSA 6) by the General Fisheries Commission for the Mediterranean (GFCM), …” (l. 77-79).

2) Line 73. I would refer here to Figure 1.

Reference added (l. 80). 

3) Line 76. There are some more works in the W Med, e.g. Pires et al.,2018. https://doi.org/10.12681/mms.2006

The sentence refers specifically to the area influenced by the western part of the Balearic front, located off the coast of Northern Catalonia, while the suggested reference focuses on southern Portugal, Gulf of Cadiz, Strait of Gibraltar and Alboran Sea. After consideration, we have chosen not to include this reference here. 

4) Line 87. I would move this table to the supplementary material, refer to Fig.1. Table 1 occupies a lot of space and the level of detail offered (Lat and Longs for example) are not needed "inmediately".

The table has been moved to supplementary material.

5) Line 127-131. Besides a-priori analyses, it is always interesting to conduct a data-driven analysis (e.g. clustering/ordination methods ), without considering the three zones. in this way, you can extract spatial patterns that may not be related to your pre-conceptions. Note: I see that you later conduct NMDS..

As the reviewer points out, NMDS analysis (a clustering/ordination method) is adressed later in line 174.

6) Methods: Despite GAMS are widely used, in my view GLMs should be preferred (can use linear models that allow for curved responses) if you do not have a clear view of why your data may be non-linear in a particular way (specially if you have few data in some areas of your graph). In general, I miss the results (values) of the GAMS/NMDS analyses and a better explanation of those results. How good was this approach?

The reason for using GAM instead of GLM is twofold. Although fitting GLMs is an option, we choose fitting GAMs because it is a powerful data-driven approach allowing to determine relationships between response and explanatory variables following known functions e.g. linear, potential, exponential or even unknown functions e.g. polynomial. Therefore, instead of assuming a prior a relationship between response and explanatory variables, we let the data indicate the type of existing relationship (similarly to fitting smoothing splines). The other reason for using GAMs is related to the R library used to assess the environmental effects on the spatial ordination from the NMDS analysis. Generally, the response variable is a vector, though in this case it is a matrix. The R library used here has implemented a regression method for dealing with a matrix as a response variable, using GAM fitting. 

The significance of the relationship between the environmental fitting and the spatial ordination of the species is indicated in the Results (lines 263-264). Additionally, here we present the statistical results related to the graphic results in Figure 7, just for reviewer information. The reason why we consider these results not worthy to show in the text, is because they are only approximate statistics (Table 0) for the smooth terms fitted with GAMs, as provided by the R library. Instead, we focused on how the environmental gradients where influencing the spatial ordination of the species. 

Table 0. Statistical results for the relationship between environmental variables and spatial ordination of main species shown in Figure 7. 

 sst sal chl

Significance of Intercept <2e-16 <2e-16 <2e-16

Approximate significance of smooth term 0.05 <0.01 <0.01

Deviance explained (%) 4.6% 12% 14%

n 101 97 100

Estimated degree of freedom of the smooth term 1.243 0.9232 3.318

Nevertheless, for clarity, we have slightly modified the results description related to Figure 7 (lines 263-268), as follows:

“The species ordination was significantly correlated linked to CTD sea surface temperature, salinity and fluorescence (p = 0.049, p = 0.001 and p = 0.003, respectively; Figure 7). A. antennatus larvae were associated with colder (Figure 7A) and more saline (Figure 7B) waters around 24 ºC and 38 in the Northern zone, while P. longirostris, S. carinata, and A. nitescens were mainly associated with low-salinity (Figure 7B) and low-fluorescence (Figure 7C) waters in the Central and Southern zones (37.8 and 0.046 µg/L).

We also have modified the legend of Figure 7 as follows: “NMDS species ordination over contours, representing environmental gradients for A: sea surface temperature (ºC), B: sea surface salinity (PSU), C: sea surface fluorescence (µg/L)…”

7) Line 175. In general, in all tests I'd include the statistical test used, its value, etc., unless not required by the journal

We have now included the statistical test and its value in all tests of this section (l. 210-220).

8) Line 178. These p values are very marginal, If the analysis was bayesian, probably you would not claim differences to be too credible. I'd be cautious with these interpretations. Later I saw you include one sentence on this in the Discussion, but still, you expand on these results.

The referred part of the discussion has been toned down and now only refers to zooplankton volume being higher in the Northern submarine canyons area (l. 283). About the general larval density, we comment: “the general decapod larval density was found to be uniform throughout the study area” (l.- 296-298).

9) Line 190. On average.

Corrected.

10) Line 214. Log10 or Ln? You mention Log10 in the Methods

It should have said Log10, the precision has been added.

11) Line 226. Again, p values tell not too much, including sample size, degrees of freedom and the test value is recommended.

As indicated above (see Reply No. 5, Table 0), we have considered not strictly necessary and not very informative to show approximate statistics of the smoother terms calculated, using the approach of regressing the species ordination vs environmental variables.

12) Line 226-229. I wonder if you are no missing some key predictive variables, such as sediment type where some adults dwell

It may very well be. Although we considered expanding the analysis to include adult habitat, time constraints led us to decide on reserving this kind of analysis for a later publication.

13) Line 251 "although followed...." that sentence is a bit difficult to understand

The sentence was in fact not well-written, it has been corrected (l. 296-298). 

14) Line 255. Do the samples from that study come from a different or the same survey?

The samples from the 2020 study come from a different survey carried out at a smaller-scale, where all stations were located within one submarine canyon (off the coast of Blanes).

Reviewer #2: 

15)The first question is related to the title and some sentences along the manuscript that from my point of view do not correspond with the real content of the work. I mean, as authors say at some point (lines 15-16), they “present the summer mesoscale larval distribution of several species of commercial interest”, as well as the objective of the work (lines 79-81): “to describe the summer community…”. But the title does not reflect this exactly, and it could create false expectations in the readers like myself. Therefore, I suggest adjusting the title to the actual content.

The title has been changed to include the reference to the period studied (summer). However, we feel like this study does provide useful information on species other than commercial ones, even if we chose to focus on those commercially exploited in the area. For this reason, we have left the general term “decapod crustacean larval communities” in the title.

16) The second question refer to the methods for collecting. Authors used a neustonic sledge, with a flowmeter to calculate the volume of filtered water. But I miss some important data about the moment of the day for collection because we know that nekton composition is not the same during the day or in the night. There are significant variations in species composition and abundances. Therefore, I think that this is an important information to understand the results. As well as the period of time that the neustonic sledge was towed. I know that finally with the flowmeter you can compare samples taken in different periods of time (from few minutes to longer), but with longer periods the probability of collecting species with lower densities increase. Therefore, to help to understand the results for readers I suggest including this information in the Method section.

Samples were collected avoiding dusk and dawn, and for this study day and night samples were treated together. The towing time was 15 minutes in all stations, which is a standard practice in plankton samplings. This has been clarified in the text (l. 105-108).

17) Respect to the area of study, authors selected three zones: Northern, Central, and Southern; but I cannot see the criterion for the limits of each one. According to figure 1, the limit between “Southern” and “Central” is not exactly the Ibiza Channel, and the limit between “Central” and “Northern” is not between Blanes and Palamos, why? Authors mention these two barriers in the Introduction (lines 66-78), so why they did not use this information to define the three zones? Instead, they define 3 zones with a clear different number of stations (Southern: 31, Central: 20, Northern: 50) and different distribution (density of stations at Northern zone clearly the highest). In the Northern zone (figure 1) is easy to see distribution of several sampling stations forming perpendicular lines to the coast line (radially), but not in the other two zones. I suggest to the authors to describe the criterion to stablish the 3 zones, and the distribution and density of stations, and how this could affect (or not) to the results and the analysis.

The reviewer is right about the unclear and redundant description of the three areas in the Introduction and in the Data Analysis sections. We have modified the two paragraphs in the manuscript to avoid confusions. We also have removed the expression “barrier” as it really refers to a density front that changes with seasons and years and is more a characteristic of the northern zone, than a boundary itself. 

In the Introduction (l. 77-85) we have removed the reference to the specific three zones to be further described in the Data analysis section:

“Moreover, a hydrodynamic model run in GSA 6 showed the area show diverse circulation patterns affecting could be divided in three different zones regarding the dispersal patterns of particles, and particularly larvae of A. antennatus (Clavel-Henry et al. 2020, Carreton et al., in press). These three area zones is mainly influenced by are defined by two hydrographic characteristics barriers: related to the western part of the Northern Current North Balearic Front, situated off the coast between Blanes and Palamós (López-García et al. 1994), which has been found to transport larvae southwestward influence decapod larvae dispersal (Clavel-Henry et al. 2021), whereas that its interaction with the Balearic current Ibiza Channel, off the coast of Dènia, where the may generate ion of seasonal eddies restricting conditioning the can enable or prevent the dispersal of larvae or transporting them offshore (García-Lafuente et al. 1995; Clavel-Henry et al. 2021).”

We have rewritten the Data Analysis description emphasizing on the differences between zones, as follows (l. 141 – 153):

“For the spatial analysis, the studied area (GSA6) was divided into three zones according to environmental characteristics such as temperature, salinity, bathymetry and surface circulation, influencing the larval dispersal, as mentioned in previous studies in the area (Clavel-Henry et al. 2020, Pastor-Prieto et al., 2021). The Northern zone englobes the deep submarine canyons from the Gulf of Lions to the coast of Barcelona with a strong effect of the saltier and cooler Northern current flowing southwest. The Central zone, includes the wide platform off the coast of Tarragona down to the Gulf of Valencia, characterized by the inshore influence of the Northern current while offshore is influenced by a countercurrent flowing northeastward, the Balearic Current, which creates a number of mesoscale circulation structures (eddies). Finally, the Southern zone, includes the Ibiza Channel and the coast off Alicante down to Cartagena (Fig. 1), is mostly influenced by less salty Atlantic water recently entering the Mediterranean through the Strait of Gibraltar and flowing northeastward.”

Regarding the perpendicular distribution of the sampling stations in the northern area, it is related to the stepped topography linked to the submarine canyons, requiring higher sampling frequencies along and outside the canyon’s axes. 

18) I know that is impossible to take samples at 101 stations along hundreds of miles simultaneity, but between the first samples and the last there are more than a month. This affects the results in two ways, different reproductive periods or pike of larval emission depending on the species could be not take into account, and these samples taken in this period do not represent “the “summer” distribution of…”. Summer start one month before starting the sampling and finish a month later. I know that authors mention this in the Discussion section (lines 258-312). But, for this reason some sentences must be corrected considering this. For example: lines 236-237, and 313-314,

Corrected in l. 260: The sentence now reads “For spatial representation of the summer decapod larval community...”. L. 359-360: “The mesoscale approach of this study allows for a broad view of the summer decapod larval community of the Eastern Spanish Mediterranean coast.”

19) I am not sure whether this is a rule of the journal, but in any case, I do not understand why all scientific names of species in the references are not in italics (see lines 338, 345, 346, 351, 354, 369, 378, 380, etc…)? The journal names are in italics, so is not a problem with this letter format.

This was due to a formatting error on our part. The references list is now updated and in the correct format.

20) Other minor comments and corrections as follow:

Line 10. Authors mention the taxa identified at the lowest level (shrimp, sensu lato), and those left at infraorder, but I miss two infraorder well-represented in marine waters, Astacidea and Polychelida, and my question is, are not mentioned because are not considered or because no larvae of these taxa were found? Absence of species is also an interesting information. For example, an Astacidea, like Nephrops norvegicus is also an commercial important species, and their larvae are extremely difficult to find in plankton samples.

No larvae of Astacidea or Polychelida were found in our samples. This sentence has been added to the text (line 11). As for the larvae of Nephrops norvegicus in particular, which have indeed been very difficult to find in the plankton, the species has its peak reproductive period in the winter months, while our sampling only covered July and August.

Lines 26-27. Probably this first sentence must be supported by a reference.

Reference added (l. 26-27): Briones-Fourzán, P.; Hendrickx, M.E. Ecology and Diversity of Marine Decapod Crustaceans. Diversity 2022,14, 614. https://doi.org/10.3390/d14080614

Lines 50-51. “the eggs hatch directly into the first zoeal stage”. This is not right. There are exceptions, species with abbreviated development, or direct development. In these cases they can hatch as decapodite/megalopa (not zoea) or even as first crab/juvenile. Correct accordingly.

Corrected: “... the eggs can hatch directly into the first zoeal stage, which undergoes several metamorphoses until it reaches the final decapodid stage that preceeds the juvenile life phase (Martin et al. 2014). In some cases, larvae undergo abbreviated or direct development and omit some of these phases (Martin et al. 2014).”. (l. 51-55)

Lines 53-65. In this paragraph I suggest citing works like:

Brandao, M. C., Freire, A. S. & Burton, R. S. (2016) Estimating diversity of crabs (Decapoda: Brachyura) in a no-take marine protected area of the SW Atlantic coast through DNA barcoding of larvae. Syst. Biodivers. 14(3), 288–302.

Marco-Herrero, E., Cuesta, J.A. & González-Gordillo, J.I. (2021) DNA barcoding allows identification of undescribed crab megalopas from the open sea. Scientific Reports 11:1.

To show the use of molecular techniques in larval studies.

Added: “Molecular techniques such as DNA barcoding are growingly used to aid in the identification process of decapod crustacean larvae (Brandao et al. 2016, Carreton et al. 2019, Marco-Herrero et al. 2021).” (l. 66-68).

Lines 66-78.

In this paragraph I suggest to cting this work, if after read it you consider that allow to explain the barriers in the studied area:

Ojeda, V., Serra, B., Lagares, C. et al. (2022) Interannual fluctuations in connectivity among crab populations (Liocarcinus depurator) along the Atlantic-Mediterranean transition. Scientific Reports, 12, 9797.

Reference added (l.76).

Line 71. I suggest that first time cited in the text you mention it in full, for people that unknow the meaning of “GSA 6”.

An explanation has been added: “... this area, designed as the fisheries management unit Geographical Subarea 6 (or GSA 6) by the General Fisheries Commission for the Mediterranean (GFCM)...” (l.76-77).

Line 73. “Carreton et al., in press”, I did not find this citation in references. I understand that papers in press can be cited and included in the reference list.

The cited paper has since been published and the complete reference has been added: (l. 81).

Lines 66-78. Considering all the information of this paragraph, I suggest including these barriers in the figures 1, 2, 5 and 6. (see for example figure 1 in Ojeda et al. (2022))

In this paragraph, our intention was to point out the environmental conditions governing the study area, but we did not go deep into the role that e.g. surface circulation may play in larval dispersal. We consider this is not the scope of this work, focused mainly on the community structure and the role played by the environmental conditions in already well defined subareas (North, center, south) with specific characteristics addressed in studies specifically dedicated to this topic. For these reasons, we have preferred to not include the lines in the figures.

Lines 104-106. Here, and in the paragraphs 191-193 and 289-294, authors explain that used molecular identification for three larvae. I miss a little bite of more information. What genes were used? COI? 16S? other? Authors say “following the method detailed by Carreton et al. (2019)”, in that work they used different primers, for different species, so I am not suggesting to include here all that information about primers and pcr protocols, but at least the gene or genes used, and the accession codes to Genbank (or BOLD, if COI) of the sequences obtained.

In this case, we used COI (added in the text in l. 125) with the primer pair CrustDF1 and HCO-2198. The GenBank accession numbers are now included in the text (l. 126-127). 

Lines 154-156. As commented above, I suggest mentioning the remaining Infraorders (as Astacidea and Polychelida), and comment that larvae of these taxa where not found in your samples. The absence, or the negative result, is also an interesting data.

Added: “No larvae of Astacidea or Polychelida were found.” (l. 191)

Lines 156-157. In fig. 2A I can see high volume of zooplankton in the Central zone too, especially in the Ebro Delta area.

Modified: “Higher values of estimated zooplankton volume were found in the Northern zone and the Ebre Delta” (l. 192).

Lines 204-207. In this analysis of non-commercial species, there are a comparison of Sergestidae (higher abundance) with the rest of taxa, but in the order of more to less represented authors forget to mention S. carinata (Fig. 5A). In this comparison, they use species (S. carinata, Alpheus glaber, Athanas nitescens), genus (Lysmata, Processa, Upogebia) and family level (Sergestidae), but in Table 3, we can see the level of identification of these taxa in families too. So, maybe could be all compared at this level? In some cases, for example S. carinata, is the same, but in other as Alpheidae would be the sum of Alpheus glaber and Athanas nitescens. It is just a suggestion.

The comment on Figure 5 was completed: “… the mud shrimp Upogebia (Fig. 5G), and the Mediterranean rock shrimp Sicyonia carinata (Fig. 5A)” (l. 245-246). Since the table format already allows for general sums by groups and comparison between different taxonomic levels, we chose to use the lowest possible taxonomic level in the distribution map of Figure 5, mainly to keep the coherence with the commercial species twin figure (Figure 6), where species level does give more relevance to the results.

Lines 236-237. Insert in this sentence the temporal window: “summer”.

Corrected.

Line 352. Include the name of the journal (PLOS ONE, I think), and volume.

Corrected: Clavel-Henry M, Solé J, Kristiansen T, Bahamon N, Rotllant G, et al. (2020) Modeled buoyancy of eggs and larvae of the deep-sea shrimp Aristeus antennatus (Crustacea: Decapoda) in the northwestern Mediterranean Sea. PLOS ONE 15(1): e0223396. https://doi.org/10.1371/journal.pone.0223396

Line 387. Complete this reference with number of Leaflet and number of pages.

Corrected: Dos Santos, A., & Lindley, J. A. (2001). Crustacea Decapoda: Larvae II. Dendrobranchiata (Aristeidae, Benthesicymidae, Penaeidae, Solenoceridae, Sicyonidae, Sergestidae, and Luciferidae). ICES Identification Leaflets for Plankton. Fiches d'identification du plancton 186.

Lines 401-406. I unknow the rules for references of this journal, but I do not understand the order of the three papers by Fuste. First 1987, second 1989 and third 1982.

Corrected: first 1989, then 1987 and lastly 1982.

Line 404. Complete this reference with number of volume and number of pages.

Corrected: Fusté, X. (1989). Distribution of Decapod Crustacean larvae in the Bays of the Ebro Delta. Scientia Marina 53(4): 763-770.

Line 406. Complete this reference with number of volume and number of pages.

Corrected: Fusté, X. (1982). Ciclo anual de las larvas de Crustáceos Decápodos de la costa de Barcelona. Investigacion Pesquera 46(2): 287-303.

Lines 432-434. This is a PhD. thesis? I suggest completing the information, University? Number of pages? Published (printed or online), unpublished?

Corrected: Guijarro, B. (2012). Population dynamics and assessment of exploited deep water decapods off Balearic Islands (western Mediterranean): from single to multi-species approach. Ph. D. Thesis, Universitat de les Illes Balears, 264 pp. Retrieved from: http://hdl.handle.net/11201/2504

Lines 442-444. This is also a PhD. Thesis? I suggest the same that above.

Corrected: Landeira, J. M. (2010). Larvas planctónicas de crustáceos decápodos en las Islas Canarias. Ph. D. Thesis, Universidad de La Laguna, 190 pp. Retrieved from papers2://publication/uuid/D7F09E6C-2BC1-4A85-BEFF-5DA19CDB75D1

Lines 455-456. This reference is confused. This is a chapter of book, so I miss data about editorial, country, pages, etc., and what about (Anger, 2001)?? Please, correct accordingly.

Corrected: Martin JW, Criales MM, Dos Santos A. 2014. Dendrobranchiata. In: Martin JW, Olesen J, Hoeg JT, eds. Atlas of Crustacean Larvae. Baltimore: Johns Hopkins University Press, 235–246.

Lines 480-483. Like above, this is a book chapter? I miss data.

Corrected: Queiroga, H., & Blanton, J. (2005). Interactions between behaviour and physical forcing in the control of horizontal transport of decapod crustacean larvae. Advances in marine biology, 47, 107–214. https://doi.org/10.1016/S0065-2881(04)47002-3

Lines 498-500. I think that the name of the journal must be included, and what mean (Lc)?

Corrected: Sobrino, I., & García, T. (2007). Reproductive aspects of the rose shrimp Parapenaeus longirostris ( Lucas , 1846 ) in the Gulf of Cadiz ( southwestern Iberian Peninsula ), 23(Lc), 57–71.Boletín Instituto Español de Oceanografía 23 (1-4). 2007: 57-71.

---

## [Decision Letter · Decision Letter 1]

26 Sep 2022

Summer decapod crustacean larval communities along the eastern Spanish Mediterranean coast

PONE-D-22-14466R1

Dear Dr. CLARET,

We’re pleased to inform you that your manuscript has been judged scientifically suitable for publication and will be formally accepted for publication once it meets all outstanding technical requirements.

Kind regards,

Antonio Medina Guerrero, Ph.D.

Academic Editor

PLOS ONE

Additional Editor Comments (optional):

Reviewers' comments:

Reviewer's Responses to Questions

**Comments to the Author**

1. If the authors have adequately addressed your comments raised in a previous round of review and you feel that this manuscript is now acceptable for publication, you may indicate that here to bypass the “Comments to the Author” section, enter your conflict of interest statement in the “Confidential to Editor” section, and submit your "Accept" recommendation.

Reviewer #1: All comments have been addressed

Reviewer #2: All comments have been addressed

2. Is the manuscript technically sound, and do the data support the conclusions?

Reviewer #1: (No Response)

Reviewer #2: Yes

3. Has the statistical analysis been performed appropriately and rigorously? 

Reviewer #1: (No Response)

Reviewer #2: Yes

4. Have the authors made all data underlying the findings in their manuscript fully available?

Reviewer #1: (No Response)

Reviewer #2: Yes

5. Is the manuscript presented in an intelligible fashion and written in standard English?

Reviewer #1: (No Response)

Reviewer #2: Yes

6. Review Comments to the Author

Reviewer #1: (No Response)

Reviewer #2: I have read the revised version and I think that it is clearly improved. Authors followed almost all suggestions, and I agree with their response in the few cases where they did not the suggested changes.

7. PLOS authors have the option to publish the peer review history of their article (what does this mean?). If published, this will include your full peer review and any attached files.

Reviewer #1: No

Reviewer #2: No

---

## [Editor Report · Acceptance letter]

14 Oct 2022

PONE-D-22-14466R1 

Summer decapod crustacean larval communities along the eastern Spanish Mediterranean coast 

Dear Dr. Company:

I'm pleased to inform you that your manuscript has been deemed suitable for publication in PLOS ONE. Congratulations! Your manuscript is now with our production department. 

Kind regards, 

on behalf of

Dr. Antonio Medina Guerrero 

Academic Editor

PLOS ONE